# A Convenient Unified Model to Display the Mobile Keyhole-Mode Arc Welding Process

**Yan Li [1,2,*]**, **Chen Su [1]**, **Ling Wang [3]** and **Chuansong Wu [4]**

[1]  College of Mechanical and Transportation Engineering, China University of Petroleum-Beijing, Beijing 102249, China; 2019215343@student.cup.edu.cn

[2]  Beijing Key Laboratory of Process Fluid Filtration and Separation, China University of Petroleum-Beijing, Beijing 102249, China

[3]  Avic Jonhon Optronic Technology Co., Ltd. Luoyang 471003, China; wangling@jonhon.cn

[4]  MOE Key Lab for Liquid-Solid Structure Evolution and Materials Processing, Institute of Materials Joining, Shandong University, Jinan 250061, China; wucs@sdu.edu.cn

*  Correspondence: heartonelee@cup.edu.cn

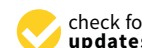

**Featured Application: A 3D unified model is developed to describe multi-physics phenomena in the mobile keyhole-mode plasma arc welding. It provides a convenient and accurate method to guide and improve the keyhole-mode plasma arc welding process.**

**Abstract:** Keyhole-mode plasma arc welding (PAW) has a good prospect in the manufacturing industry. Unified models of plasma arc and workpiece help to reveal the physical mechanism in PAW for a better application. Previous unified models either deal with a constant keyhole situation or take too much computational time to display the dynamic keyhole process with a two-phase flow method. In view of the convenience for industrial application as well as good accuracy, a convenient unified model was developed to describe the mobile keyhole-mode PAW. With a simplified technique, the multiphase heat and force effect between plasma arc and workpiece was turned into a single-phase problem at each individual domain. Thus, it takes less computational time than previous unified models. The temperature field and weld pool during the mobile keyhole-mode PAW process were revealed, the arc flow and pool flow were displayed and the electric potential was predicted. The experiment was conducted on a stainless-steel plate, and the weld pool image and the measured arc pressure agree well with the calculated result. The calculated electric potential drop also coincides with the experiment. The model provides a convenient and accurate method to display the mobile keyhole-mode arc welding process.

**Keywords:** unified arc welding model; keyhole-mode welding process; weld pool; heat transfer; multi-physics; electric potential

## 1. Introduction

Arc welding is widely used in the manufacturing of industrial equipment, buildings, bridges, vehicles, turbines and pressure vessels. However, good groove preparation, proper filler wire and multi-layer and multi-pass welding are usually necessary with traditional arc welding technologies. More efficient and advanced welding technologies are demanded for modern equipment and sophisticated equipment. Recently, high-power welding technologies have attracted a lot of attention because they are available to produce qualified weld beads with a single pass, and even need no groove preparation. As one of the high-power welding technologies, plasma arc welding (PAW) is superior to others due to lower equipment costs, convenient operation and higher tolerance to joint gaps [1].

The arc is concentrated by a nozzle in PAW, so the heat flux and arc pressure are much higher than that in ordinary arc welding. As a consequence, it can quickly produce a weld pool and then penetrate through the pool, developing into a complete keyhole full of arc. Figure 1 displays a keyhole-mode PAW process. As the arc temperature is very high, much heat can be directly transported to the lower part through the keyhole, which improves the welding penetration. In this way, it can avoid groove preparation and weld thicker pieces with a single pass.

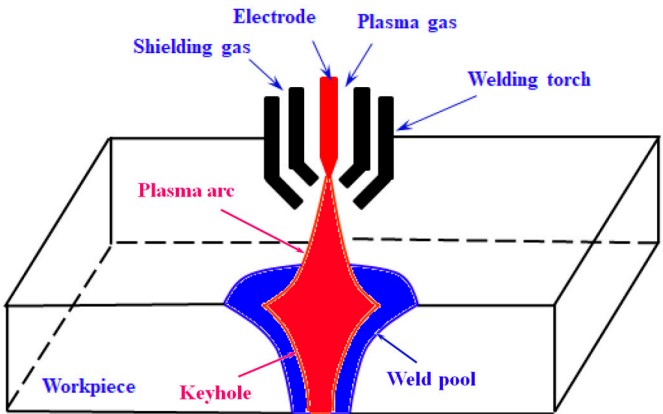

**Figure 1.** A keyhole-mode plasma arc welding.

In view of the above advantages, researchers have carried out tremendous experimental research about PAW [2–4] to acquire the keyhole mode welding technology. Up to now, the formation and development of both weld pool and keyhole during the welding process were successfully detected [5,6]. However, PAW also has some disadvantages. For example, PAW does not use the AC power, and welding thickness is generally no more than 15 mm at the keyhole mode. The tungsten electrode and nozzle are easily burned due to the hot arc. Especially, multiple welding parameters have an important effect on the PAW, and proper welding specifications are required to guarantee the weld quality. Experimental research just displays limited information about the welding process, while numerical calculations can help reveal full and detailed physical mechanism. Therefore, researchers developed a lot of arc welding models in recent years to simulate heat transfer, liquid metal flow and weld pool. For simplicity, some researchers neglected plasma arc in calculations but just constructed an equivalent heat source model [7,8] in workpiece to predict the temperature development and weld pool morphology. Feng et al. [9,10] developed a heat source model with a Lattice Boltzmann method (LBM) to study the weld pool behaviors in PAW. In order to reveal the keyhole effect between plasma arc and liquid metal, some researchers apply a two-phase flow model to calculate their interaction. Volume of Fluid (VOF) method, which can locate the phase interface of two or more fluids [11], is widely used to simulate the keyhole welding process. Wu et al. [12,13] used the VOF method to construct a keyhole-related heat source model, arc pressure model and shear stress model to study the formation mechanism of the weld pool and keyhole. They showed the molten flow patterns in the weld pool and indicated arc pressure and shear stress take the main responsibility for keyhole formation. Li et al. [14,15] also established an arc heat model and arc pressure model with VOF to describe heat transfer, liquid metal flow and keyhole surface evolution in PAW. They found liquid metal in the front flows bypass the keyhole wall to the back, and the keyhole channel is crocked because of the movement. Up to now, most of models need the VOF method to simulate the dynamic evolution with respect to the keyhole [16–19].

However, there is no plasma arc in all the above models, and they only take the workpiece into account so that they all rely on a reasonably designed heat source model and arc pressure model. Researchers' experience and assumptions on the shape, size and flux of the two models play a critical role to accurately display the heat and force effect in the welding process. A unified model of the plasma arc and weld pool is required to avoid these different subjective assumptions of heat source and

arc pressure. A lot of unified models of gas tungsten arc welding (GTAW) [20–22] have been developed, but there is a problem with these models in that they are not able to calculate the keyhole-mode welding phenomenon as they assumed the weld pool surface is flat, which is quite different to the keyhole phenomenon.

Just in recent years, several researchers began to construct unified models of PAW to analyze the keyhole effect. According to experimental observations, Li et al. [23] pre-set a constant keyhole and established a unified keyhole-mode PAW model to display heat transfer from the keyhole to the weld pool. Their model shows there are two opposite circular flows in the weld pool and about 10% of plasma arc outflows from the bottom keyhole. Xu et al. [24] also constructed a 2D unified model with a pre-set keyhole to study physical characteristics of the plasma arc after full penetration. They demonstrate the plasma arc will be compressed within the keyhole since the keyhole radius gradually decreases. Wang et al. [25] displayed heat transfer and fluid flow in a unified GTAW model, in which a constant keyhole was pre-set based on experimental observation. Wu et al. [26] developed a 3D PAW model to analyze the energy propagation mechanism in the keyhole-mode welding process when the weld pool was at a quasi-steady state. Meanwhile, a pre-constructed keyhole is also required in their model. The above models managed to show some physical mechanisms in the keyhole-mode welding process, but they can only show the phenomena after keyhole formation and with a stable keyhole. Jian et al. [27,28] established a unified model with VOF to display the whole welding process from arc ignition to open keyhole formation in PAW. They demonstrated that the arc shape, temperature profile and current density distribution all vary dynamically as the keyhole increases. However, the predicted open keyhole time is shorter than the experiment. Pan et al. [29] proposed a model including tungsten, plasma and an anode to analyze the heat transfer and fluid flow in a keyhole PAW process. They also used the VOF to display the dynamic keyhole process and calculated energy input, transformation, transfer and dissipation in the process. However, their model uses a simplified Ampere's circumfluence law to calculate the magnetic field, which is only suitable in a 2D structure. What is more, since the VOF method is used in the above models to track the keyhole surface, it costs too much computational time, and the predicted weld pool geometry does not coincide enough with the experiment now.

In view of the convenience for industrial application as well as good accuracy, Li et al. [30] proposed a new unified model to describe the keyhole-mode heat transfer characteristics and predict weld pool geometry more conveniently. The model directly calculates heat transfer from the plasma arc to the workpiece instead of depending on assumed heat source models, meanwhile it predicts more accurate arc pressure distribution rather than empirical arc pressure models. The heat and force effect calculated from the arc region is transported to the workpiece region, and the unified model simulates a heat transfer phenomenon in each individual region. In this way, it simplifies the original two-phase problem into a single-phase problem in two different regions. So, the model takes quite less computational time than the VOF-unified models, meanwhile it is theoretically more accurate than the assumed heat source models and arc pressure models. As the previous model is just suitable to a 2D spot-welding process, we extended it into a 3D mobile model in the paper. The temperature field and weld pool evolution during the mobile PAW process were revealed, the arc flow and pool flow were displayed, and the electric potential was predicted. The PAW experiment was conducted to verify the model, and it found that the measured arc pressure and the experimental weld pool both agree well with the calculated result. The calculated electric potential drop also coincides with the experiment, while previous models never show the comparison.

## 2. Mathematical Model

Figure 2 shows the calculation domain of the unified PAW model, including welding nozzle, gas region and workpiece. Argon enters the nozzle through the inlet and ionizes to be thermal plasma due to the electric voltage between the two electrodes. There is a shielding gas around the welding nozzle, where the argon gas is also used. The plasma arc impacts on a workpiece, and it soon fuses

part of the workpiece and produces a weld pool. At the same time, the arc causes a huge stagnation pressure on the workpiece and flows out through the upper outlet boundary. To calculate a mobile process, the workpiece is set with a welding speed while the welding torch maintains a still stance.

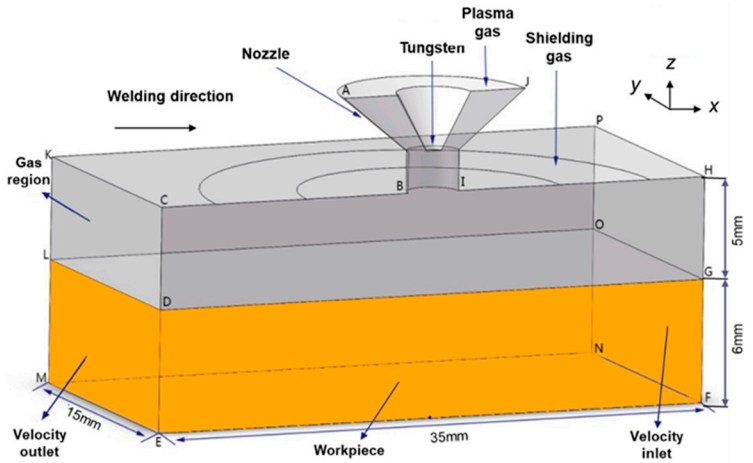

**Figure 2.** Calculation domain of the unified plasma arc welding (PAW) model.

Some assumptions are made to construct the governing equations: the plasma arc is in local thermo-dynamic equilibrium (LTE) and optically thin to radiation so that arc properties can be regarded as a function of temperature; both arc and liquid metal are in laminar flow; thermal properties of the solidus and liquidus metal can be regarded as a function of temperature; metal vapor is ignored as it has little effect on weld pool; the calculation area is symmetrical and half of the welding area is taken for calculation.

Mass conservation equation:

$$\frac{\partial \rho}{\partial t} + \frac{\partial}{\partial x}(\rho u) + \frac{\partial}{\partial y}(\rho v) + \frac{\partial}{\partial z}(\rho w) = 0 \tag{1}$$

Momentum conservation equations:

$$\frac{\partial(\rho u)}{\partial t} + u\frac{\partial}{\partial x}(\rho u) + v\frac{\partial}{\partial y}(\rho u) + w\frac{\partial}{\partial z}(\rho u) =$$
$$-\frac{\partial P}{\partial x} + \mu\left(\frac{\partial^2 u}{\partial x} + \frac{\partial^2 u}{\partial y} + \frac{\partial^2 u}{\partial z}\right) + j_y B_z - j_z B_y + S_x \tag{2}$$

$$\frac{\partial(\rho v)}{\partial t} + u\frac{\partial}{\partial x}(\rho v) + v\frac{\partial}{\partial y}(\rho v) + w\frac{\partial}{\partial z}(\rho v) =$$
$$-\frac{\partial P}{\partial y} + \mu\left(\frac{\partial^2 v}{\partial x} + \frac{\partial^2 v}{\partial y} + \frac{\partial^2 v}{\partial z}\right) + j_z B_x - j_x B_z + S_y \tag{3}$$

$$\frac{\partial(\rho w)}{\partial t} + u\frac{\partial}{\partial x}(\rho w) + v\frac{\partial}{\partial y}(\rho w) + w\frac{\partial}{\partial z}(\rho w) =$$
$$-\frac{\partial P}{\partial z} + \mu\left(\frac{\partial^2 w}{\partial x} + \frac{\partial^2 w}{\partial y} + \frac{\partial^2 w}{\partial z}\right) + j_x B_y - j_y B_x - \rho g + S_z \tag{4}$$

Energy conservation equation:

$$\frac{\partial(\rho H)}{\partial t} + u\frac{\partial(\rho H)}{\partial x} + v\frac{\partial(\rho H)}{\partial y} + w\frac{\partial(\rho H)}{\partial z}$$
$$= \frac{\partial}{\partial x}\left(\frac{k}{c_p}\frac{\partial H}{\partial x}\right) + \frac{\partial}{\partial y}\left(\frac{k}{c_p}\frac{\partial H}{\partial y}\right) + \frac{\partial}{\partial z}\left(\frac{k}{c_p}\frac{\partial H}{\partial z}\right) + S_e \tag{5}$$

Current conservation equation:

$$\nabla^2 \phi = 0 \tag{6}$$

Ohm's law:

$$\mathbf{J} = -\sigma \nabla \phi \tag{7}$$

Magnetic potential vector equations:

$$\nabla^2 \mathbf{A} = -\mu_0 \mathbf{J} \tag{8}$$

$$\mathbf{B} = \nabla \times \mathbf{A} \tag{9}$$

where $t$ is time; $\rho$ is density; $P$ is pressure; $u$, $v$, $w$, $j_x$, $j_y$, $j_z$, and $B_x$, $B_y$, $B_z$ are velocity, current density, and magnetic induction intensity in $x$, $y$, $z$ direction, respectively; $\mathbf{A}$ is magnetic potential vector; $\mu$ is dynamic viscosity; $T$ is temperature; $k$ is thermal conductivity; $\sigma$ is electrical conductivity; $c_p$ is specific heat capacity; $H$ is total enthalpy; $S_x$, $S_y$, $S_z$ are source term of momentum equations; $S_e$ is source term of energy equation; $g$ is gravity acceleration; $\phi$ is electric potential; $\mu_0$ is permeability of vacuum.

In arc region, there are only electromagnetic force and gravity, so the momentum source terms in Equations (2)–(4) are all zero. In consideration of Joule heat, electron transfer energy and radiative heat loss in the arc, the energy source term is expressed as

$$S_e = \frac{j_x{}^2 + j_y{}^2 + j_z{}^2}{\sigma} + \frac{5k_b}{2e}\left(j_x\frac{\partial T}{\partial x} + j_y\frac{\partial T}{\partial y} + j_z\frac{\partial T}{\partial z}\right) - S_r \tag{10}$$

where $k_b$ is the Boltzmann constant; $e$ is electronic charge; $S_r$ is radiant heat loss.

In the workpiece region, the Enthalpy-porous technique [31] is applied to deal with the melting and solidification problem. It is considered that there is a solidus line and a liquidus line when the workpiece is fused. The zone between the two-phase line is considered full of porous medium composed of both liquid phase and solidus phase. The liquidus volume fraction is expressed as

$$f_l = \begin{cases} 0 & T \leq T_s \\ (T - T_s)/(T_l - T_s) & T_s < T < T_l \\ 1 & T \geq T_l \end{cases} \tag{11}$$

where $T_s$ is solidus temperature; $T_l$ is liquidus temperature.

Darcy damping force takes effect in the porous zone, and a buoyancy force also appears due to temperature differences in the whole weld pool, so the momentum source terms are expressed as

$$S_x = -\frac{\mu_l}{K}u, \quad S_y = -\frac{\mu_l}{K}v, \quad S_z = -\frac{\mu_l}{K}w + \rho g\beta(T - T_m). \tag{12}$$

$$K = \frac{f_l^3}{A_0(1 - f_l)^2} \tag{13}$$

where $\mu_l$ is dynamic viscosity of liquid mental; $\beta$ is thermal expansion coefficient; $T_m$ is melting temperature; $K$ is permeability; $A_0$ is a morphology coefficient related to the porous zone.

Based on the Enthalpy-porous technique, total enthalpy $H$ is expressed as

$$H = h + f_l L = \int_{T_0}^{T} c_p dT + f_l L \tag{14}$$

where $h$ is sensible enthalpy; $L$ is latent heat.

Special treatment is needed at the interface of plasma arc and workpiece because a thin non-LTE sheath appears there, and the physical mechanism is too complex and still unknown. The "LTE-diffusion approximation" method [32] is generally applied to deal with the heat transfer problem from arc to workpiece. It neglects the real sheath but sets mesh size at the interface approximate to the diffusion length of electrons. In this way, a heat source term, which includes heat conduction, electron

condensation and radiative heat loss, is simply added to the interface. As heat conduction is calculated through the coupled interface, then the other terms should be included in the heat source.

$$S = |j|\varphi_w - \varepsilon\alpha T^4 \tag{15}$$

where $j$ is current density; $\varphi_w$ is work function of workpiece; $\varepsilon$ is emissivity; $\alpha$ is the Stefan–Boltzmann constant.

Figure 3 shows all the driving forces in weld pool. Plasma arc pressure $p_{arc}$ impacts vertically on the top surface, and it exhibits a normal distribution with a maximum pressure $P_{max}$ at the center and a sharp reduce within a radius $r_0$. The pressure can be obtained according to the governing equations in the arc region.

$$p_{arc} = P_{max}\exp(-3r^2/r_0^2) \tag{16}$$

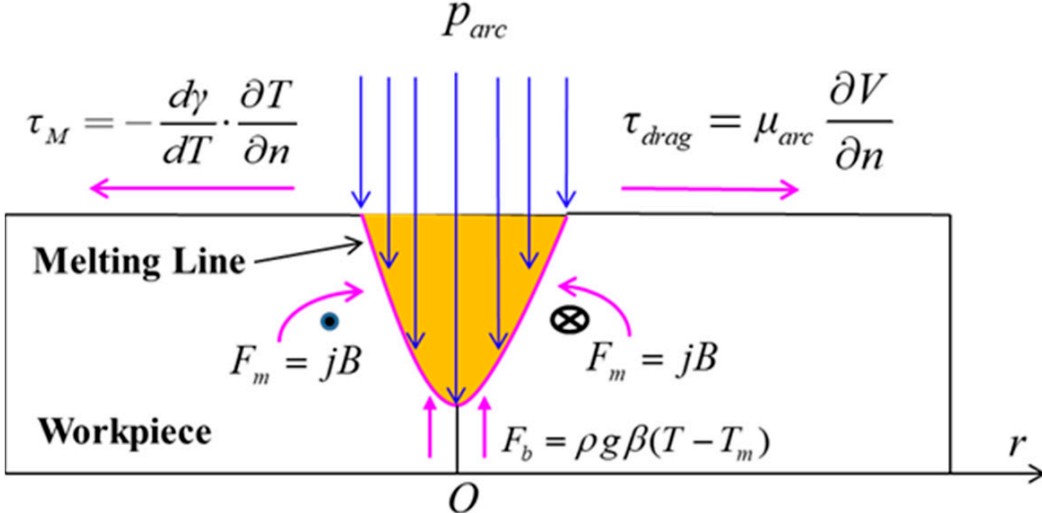

**Figure 3.** All the driving forces in weld pool.

When a keyhole appears, the arc fills up the whole keyhole. It is found that the front keyhole wall almost coincides with the solid–liquid interface when the plasma arc is penetrating through the workpiece. Therefore, as a simplification of arc force, all the melted zone is assumed full of arc pressure, and it gradually grows in the whole process. Plasma arc drag force $\tau_{drag}$, Marangoni shear $\tau_M$, electromagnetic force $F_m$ and buoyance force $F_b$ are all taken into account in the model. As shown in Figure 3, plasma arc drag force and Marangoni shear both drive liquid metal outwards, while the electromagnetic force pushes the liquid metal inwards, and the buoyance force pushes the liquid metal upwards.

The boundaries and sizes have been shown in Figure 2. There are two velocity inlet boundaries at the top. One is for plasma gas and the other is for shielding gas. The model consists of two layers, in which the top layer is gas region whose side boundaries are all pressure outlet boundary. The bottom layer is workpiece endowed with a velocity of welding speed. The right side and left side of the workpiece are set as velocity inlet boundary and velocity outlet boundary, respectively. Both velocities are equal to the welding speed. The longitudinal section ABCDEFGHIJKLMN ($y = 0$) is a symmetrical plane, and other boundaries are solid wall. The boundary conditions are displayed in Table 1.

**Table 1.** Boundary conditions of the unified model.

| Boundary | *V* (m/s) | *T* (K) | $\phi$ (V) | *A* (T•m) |
|---|---|---|---|---|
| Plasma gas inlet | 3 L/min | 300 | $\partial\phi/\partial n = 0$ | $\partial A/\partial n = 0$ |
| Shielding gas inlet | 15 L/min | 300 | $\partial\phi/\partial n = 0$ | $\partial A/\partial n = 0$ |
| Pressure outlet | — | 300 | $\partial\phi/\partial n = 0$ | $\partial A/\partial n = 0$ |
| Tungsten tip | — | 3000 | $j = I/\pi r^2$ | $\partial A/\partial n = 0$ |
| Nozzle wall | — | 1000 | $\partial\phi/\partial n = 0$ | $\partial A/\partial n = 0$ |
| Workpiece inlet | −120 mm/min | 300 | 0 | 0 |
| Workpiece outlet | −120 mm/min | 500 | 0 | 0 |
| Side workpiece wall | — | $h_{conv}\Delta T + \varepsilon\alpha\left(T^4 - T_\infty^4\right)$ | 0 | 0 |
| Top surface | — | 300 | $\partial\phi/\partial n = 0$ | $\partial A/\partial n = 0$ |
| Bottom surface | — | 300 | 0 | 0 |

## 3. Numerical Calculation

The computational fluid dynamics software Fluent was used to calculate the unified model. The computational domain is firstly discretized by a very dense and unstructured grid system. To ensure an accurate calculation of the electric-magnetic-thermal-hydrodynamical effect in plasma arc, grid size within the torch is the smallest, which is just 0.1 mm. A smaller grid size is applied in the area beneath the welding torch, and the size is 0.2 mm. In the front of the torch, a relatively bigger grid size is used, while behind the torch, a very big grid size is applied. In this way, both the computational accuracy and the efficiency are considered. Figure 4 shows the detailed grid system of the model.

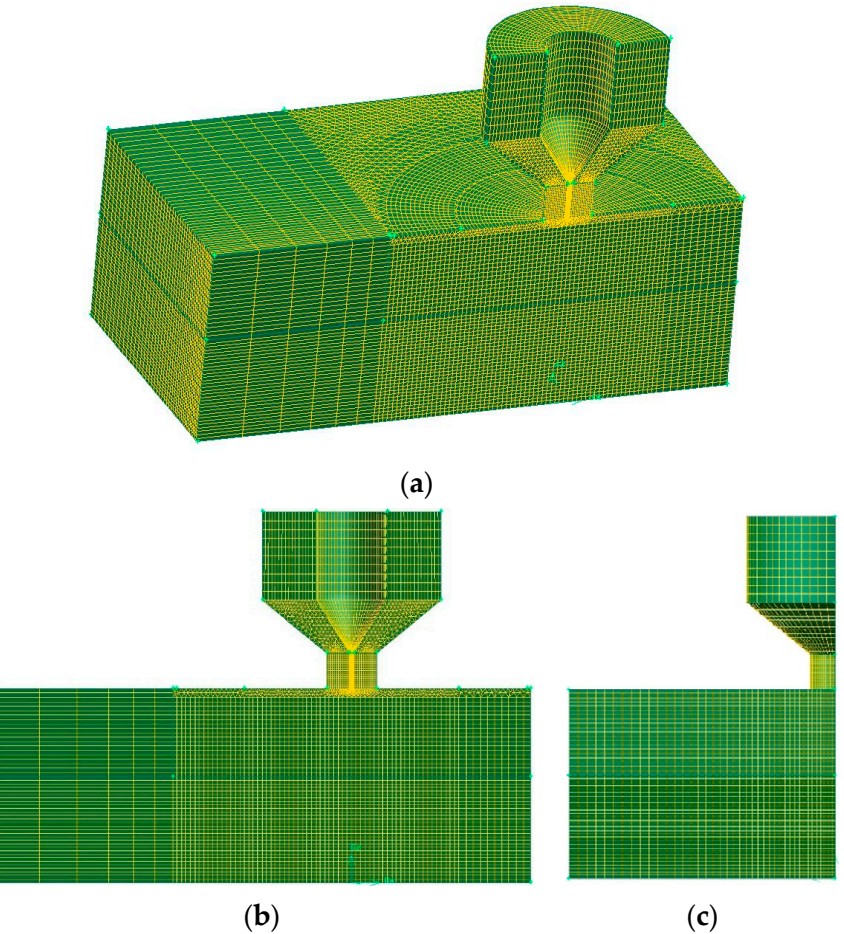

**(a)**

**(b)**          **(c)**

**Figure 4.** Grid system of the unified model. (**a**) Overall grid map; (**b**) Front-view grid (*x-z* plane); (**c**) Side-view grid (*y-z* plane).

The current conservation equation and magnetic potential vector equation were established by the User-defined Scalars (UDS) module in Fluent. All the source terms and physical parameters, such as magnetic induction intensity, current density, Joule heat, electron transfer energy, radiative heat loss, electromagnetic force, plasma arc drag force, Marangoni force and buoyancy force, are calculated by a self-developed code with the User-Defined Function (UDF) module in Fluent.

The governing equations are solved by the Pressure-Implicit with Splitting of Operators (PISO). Momentum conservation equations, energy conservation equation and UDS equations are all calculated with the second-order upwind discrete scheme. A small time step $10^{-3}$ s was used in calculation, and the sum of the normalized residues for the mass and momentum equations, energy equation and UDS equations is set to be below $10^{-3}$, $10^{-6}$ and $10^{-4}$, respectively.

Both plasma gas and shielding gas are pure argon in the PAW process, and the thermophysical properties are only a function of temperature. All the parameters used in the model can refer to literature [30]. The workpiece is stainless-steel 304, and its thermophysical properties such as density, specific heat, viscosity and thermal conductivity change with temperature, as shown in reference [8]. The process parameters are all set equal to the experimental conditions as in Part 4.

## 4. Results and Discussion

Figure 5 displays temperature field in the mobile PAW process. Arc is restricted by the nozzle and isotherms are very dense within it. The arc then spreads outwards into a bell shape and impacts vertically on the workpiece. A high-temperature area appears below the electrode, which is like an arc column located in the center of the arc. The highest temperature is above 27,000 K adjacent to the electrode tip, while the lowest temperature is 15,000 K at the edge of the arc column. Workpiece beneath the welding torch absorbs more heat from the plasma arc and quickly melts into a small weld pool at the center. The pool grows in width and depth during the welding process until it finally penetrates the whole workpiece. Due to the relative movement, the weld pool is not symmetrical along the welding direction. It shows that the pool is wide and deep in the front, while it is long and shallow in the rear. It is found that isotherms near the front pool edge are dense while those near the rear edge are distributed, which means the front pool edge absorbs more thermal energy and when it moves backwards the heat dissipates gradually. The torch axis is almost identical with the pool center line at first. However, a deviation appears with the relative movement, and it increases during the welding process. At 3.2 s, the deviation of the two central line reaches to 3.5 mm.

Figure 6 displays dynamic evolution of weld pool in the mobile PAW process. At first, a chestnut-like pool appears. However, there is a small corner at the rear. The pool grows continually, and the rear corner also increase simultaneously. However, it mainly spreads along the negative welding direction but has a small depth which is no more than 2 mm. Finally, the front pool penetrates the whole workpiece at 3.2 s. The upper pool is long and wide, but it gradually shrinks along the z direction, forming a boot-shape weld pool.

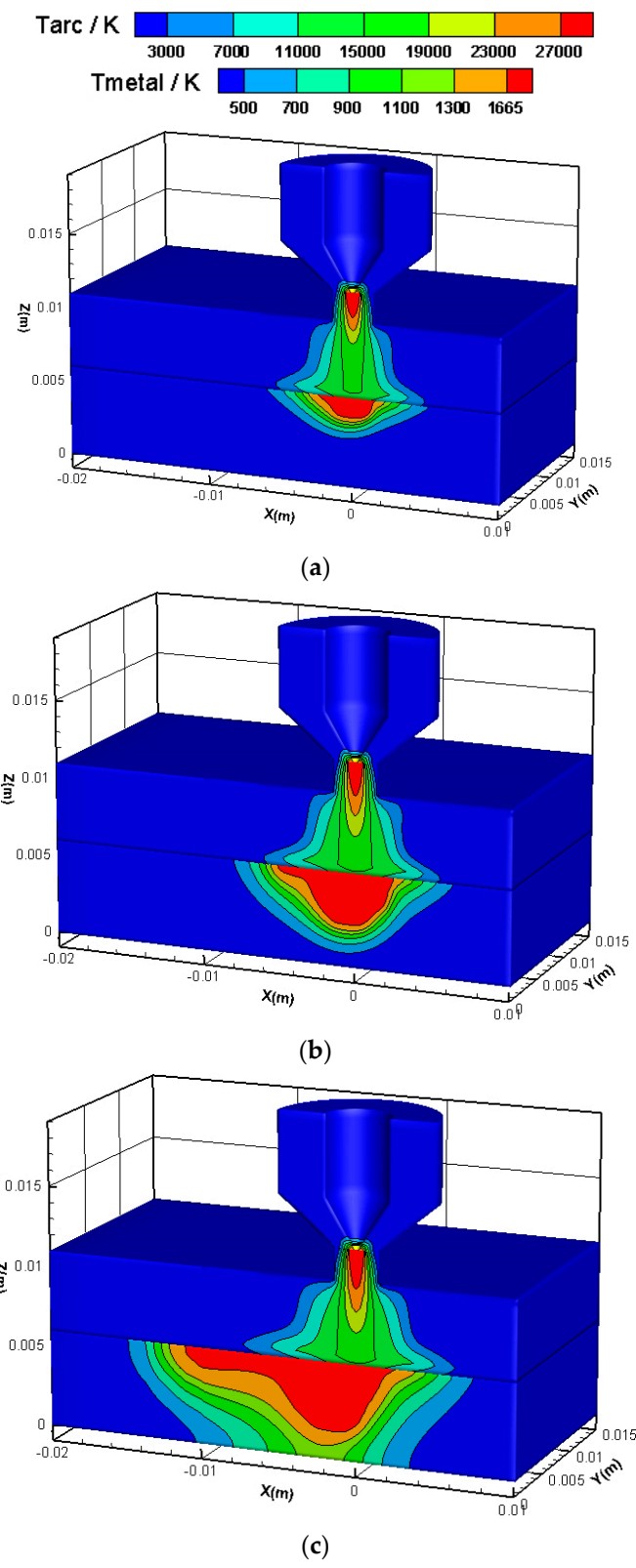

**Figure 5.** *Cont.*

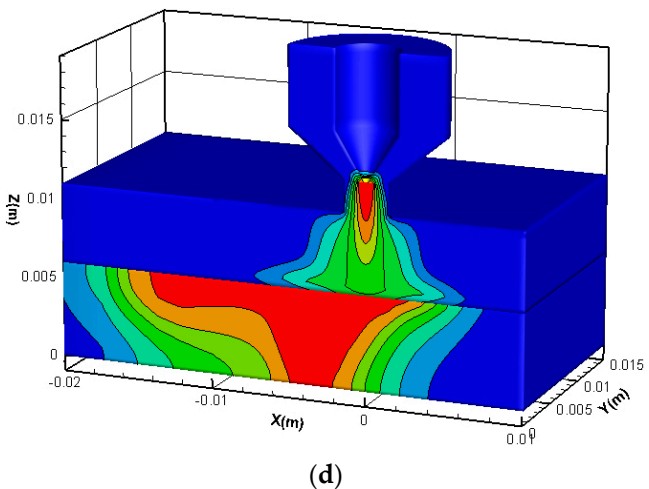

(**d**)

**Figure 5.** Temperature field in the mobile PAW process. (**a**) $t$ = 0.4 s; (**b**) $t$ = 0.9 s; (**c**) $t$ = 2.1 s; (**d**) $t$ = 3.2 s.

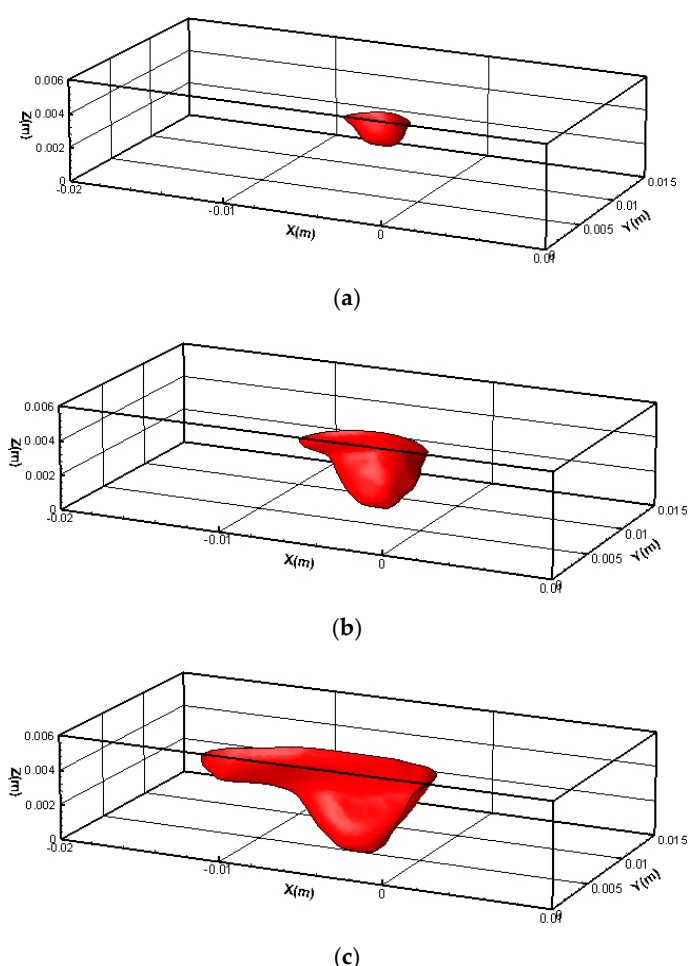

(**a**)

(**b**)

(**c**)

**Figure 6.** *Cont.*

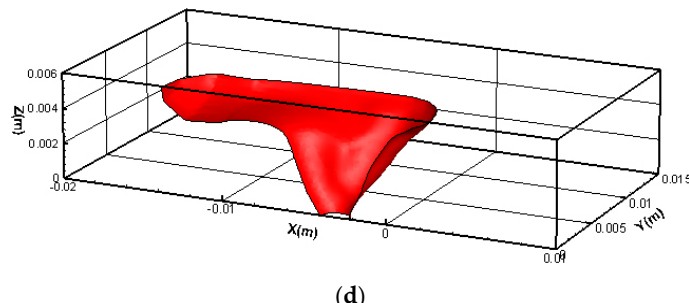

**(d)**

**Figure 6.** Dynamic evolution of weld pool in the mobile PAW process. (**a**) $t$ = 0.4 s; (**b**) $t$ = 0.9 s; (**c**) $t$ = 2.1 s; (**d**) $t$ = 3.2 s.

Figure 7 shows arc flow and liquid metal flow in the computational domain. The top displays arc velocity, where higher velocities are near the central line, and they are highly concentrated. The arc with velocity above 200 m/s almost gathers within a column of diameter of 3 mm, and that below 200 m/s diffuses at the top of workpiece, and the diameter reaches to about 10 mm. Stream traces of plasma gas show that the gas enters from top inlet, is ionized within the torch, and finally flows out around the bottom column. There is a maximum velocity zone just out of the torch, which is long and narrow, and the velocity is above 1100 m/s. Such high arc velocity brings forth very high pressure, so it promotes the liquid metal to flow downward. Then it is blocked by the front pool edge and flows backward and upward to the rear. The metal velocity is about $10^{-1}$ m/s, and the maximum velocity may reach to 0.43 m/s. References [33,34] reveal the measured average velocity of liquid metal is approximately 0.24–0.42 m/s in the keyhole-mode PAW, which basically coincides with the calculated result.

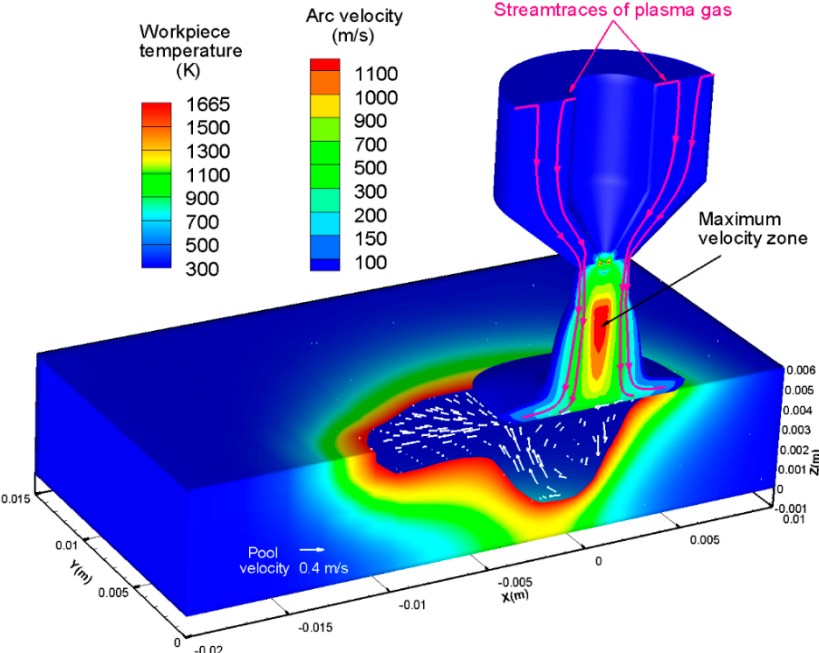

**Figure 7.** Flow distribution in the arc and weld pool.

Figure 8 displays weld pool and temperature field on the transverse cross-section (*yoz* plane) at 3.2 s. The temperature field shows the closer to the weld pool, the higher the temperature gradient. There is a maximum weld pool at $x$ = −3 mm, and the whole workpiece is fused from top to bottom. While weld pool decreases quickly in front of the cross-section. The depth reduces to 5.2 mm at $x$ = −2 mm and 0.75 mm at $x$ = 1 mm. On the contrary, the pool decreases relatively smoothly at the rear. There is still a full-penetrated weld pool at $x$ = −4 mm. Furthermore, although the depth gradually

reduces along the rear, it reaches to 1.2 mm at $x = -13$ mm. The above analysis indicates that there is a long weld pool at the rear but a narrow pool in the front.

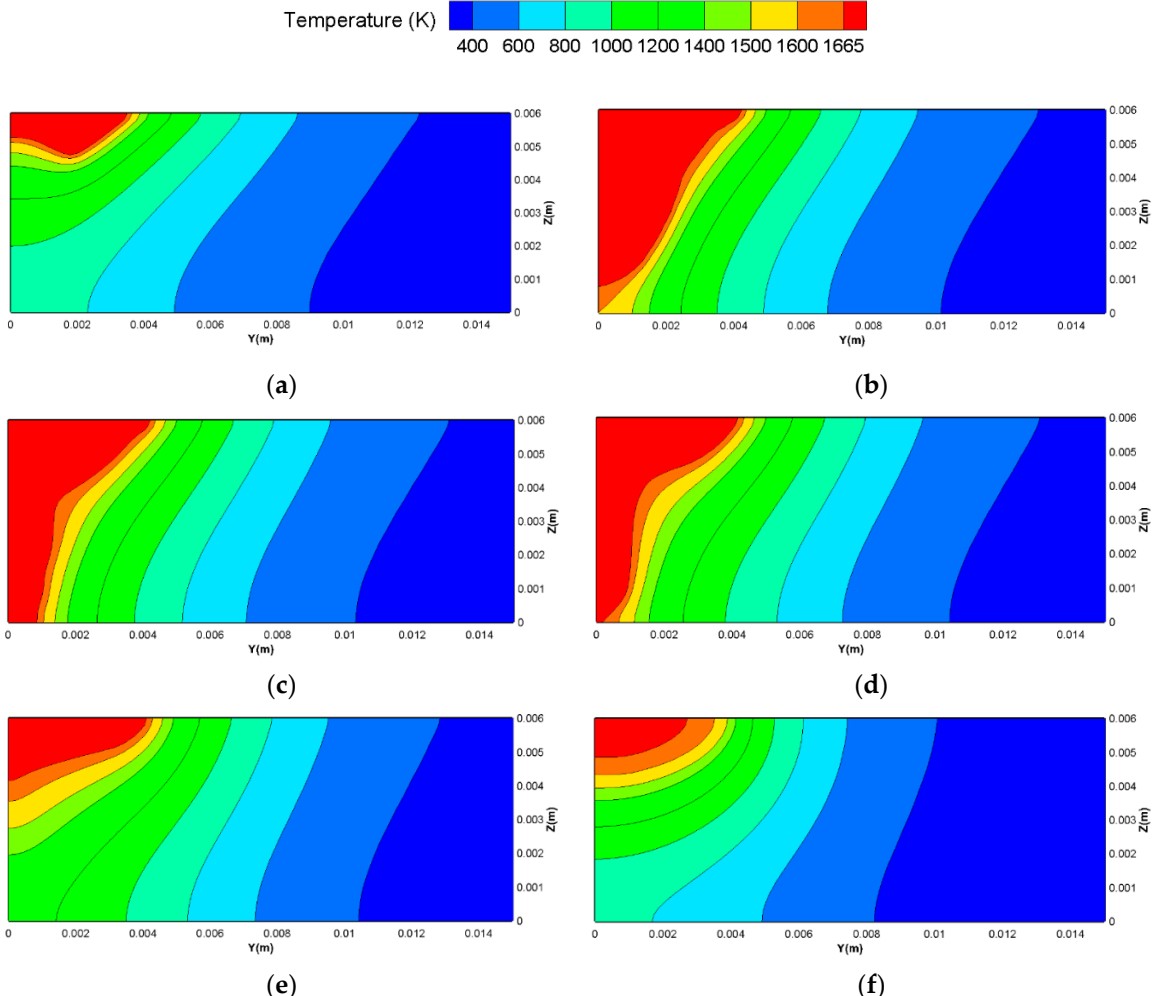

**Figure 8.** Weld pool and temperature field on the transverse cross-sections ($t = 3.2$ s). (**a**) $x = 1$ mm; (**b**) $x = -2$ mm; (**c**) $x = -3$ mm; (**d**) $x = -4$ mm; (**e**) $x = -6$ mm; (**f**) $x = -13$ mm.

Figure 9 displays weld pool and temperature field on the longitudinal section (*xoy* plane) at 3.2 s. It shows temperature field gradually shrink from the top surface to the bottom surface. The weld pool length is 16.2 mm at $z = 6$ mm, 5.6 mm at $z = 4$ mm, 3.8 mm at $z = 2$ mm, and 1.8 mm at $z = 0$ mm. The length drop rate (length reduction per unit height) reaches to 5.3 mm/mm, 0.9 mm/mm and 1.0 mm/mm in the three height ranges, respectively. The data indicates that the pool can maintain a long length at the top surface, but it shrinks fast at the upper part and smoothly at the lower part. Above research may benefit to the deep knowledge of the internal weld pool geometry.

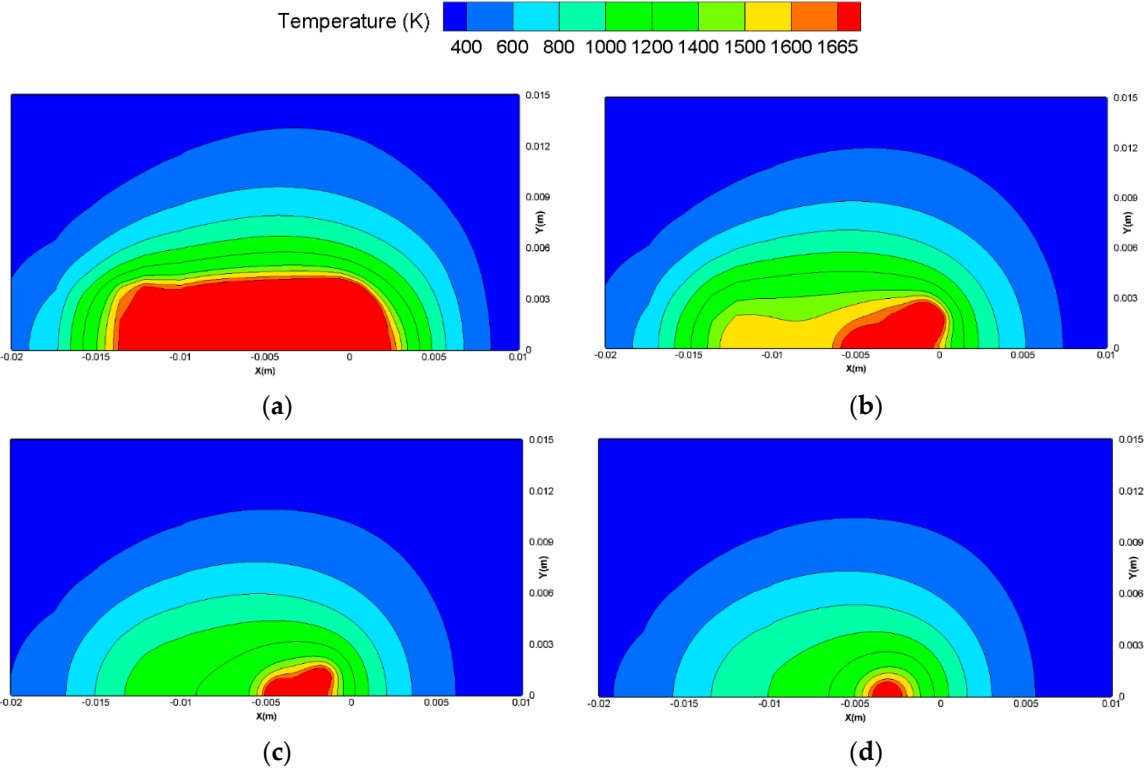

**Figure 9.** Weld pool and temperature field on the longitudinal sections (*t* = 3.2 s). (**a**) *z* = 6 mm; (**b**) *z* = 4 mm; (**c**) *z* = 2 mm; (**d**) *z* = 0 mm.

In order to verify the model, a Fronius welding machine TranSynergic 5000 was used to conduct the experiment, as displayed in Figure 10. Experimental conditions are as follows: Welding current 150 A, Welding voltage 21.5 V, Welding speed 120 mm/min, Thickness of workpiece 6 mm, Orifice diameter 2.8 mm, Plasma gas flow rate 3 L·min$^{-1}$, Shielding gas flow rate 15 L·min$^{-1}$, Tungsten electrode shrinkage 2 mm, Distance from torch to workpiece 5 mm.

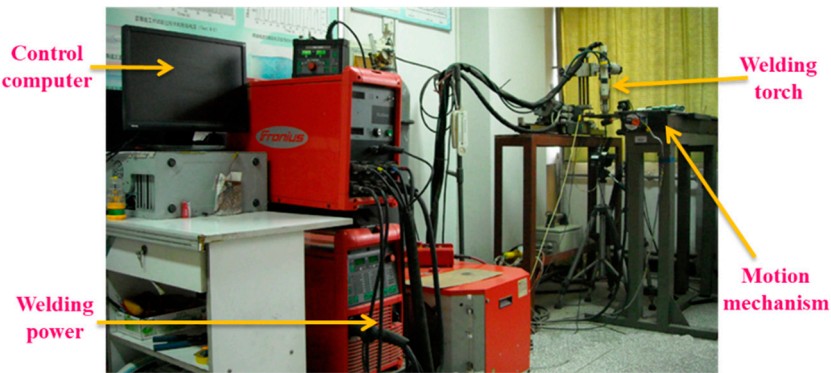

**Figure 10.** Plasma arc welding system.

Figure 11 displays comparison of captured image and calculated result of weld pool at the transverse cross-section. In the model, the solidus line (T = 1665 K) for the stainless-steel 304 is taken as the melting line. It can be found that the two agree well with each other. Figure 12 displays the calculated electric potential on the symmetry, and the maximum potential drop is 20.5 V. Meanwhile, the experimental welding voltage is 21.5 V, which is close to the calculated result. Previous PAW models never show the comparison of the welding voltage.

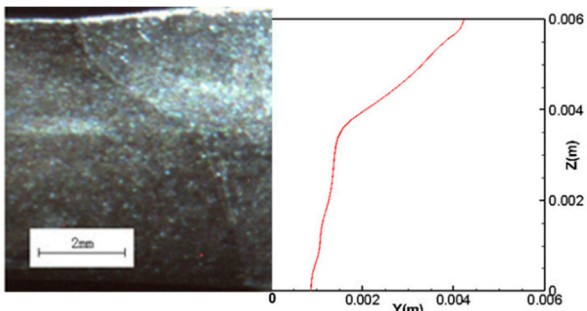

**Figure 11.** Comparison of captured image and calculated result of weld pool at the transverse cross-section.

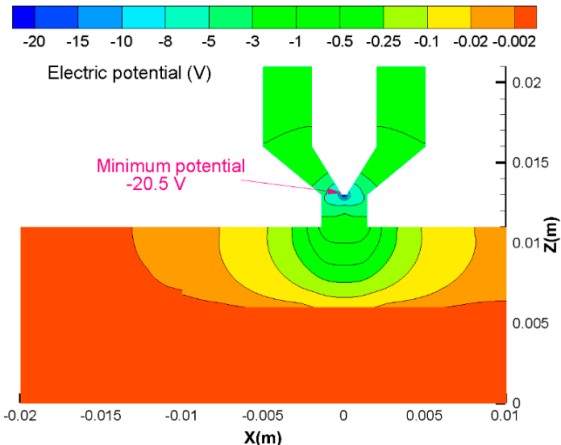

**Figure 12.** Calculated electric potential on the symmetry.

Arc pressure was also measured, and Figure 13 displays a comparison of the calculated arc pressure and measured pressure on the workpiece surface. It shows that the calculated pressure and measured pressure are in good agreement with each other at 110 A. The calculated arc pressure at 150 A shows similar distribution with that at 110 A. However, the pressure at 150 A was not measured since the detection probe will be fused at such a high current. In the unified model, the calculated arc pressure distribution at 150 A was used, where the maximum arc pressure is 6500 Pa at the center, but it reduces fast along the radial direction. The arc pressure is nearly zero beyond $r = 3$ mm. As shown in Figure 11, the $P_{max}$ is 6500 Pa, and $r_0$ is 3 mm in Equation (16). All above comparisons indicate the model is reliable in PAW engineering.

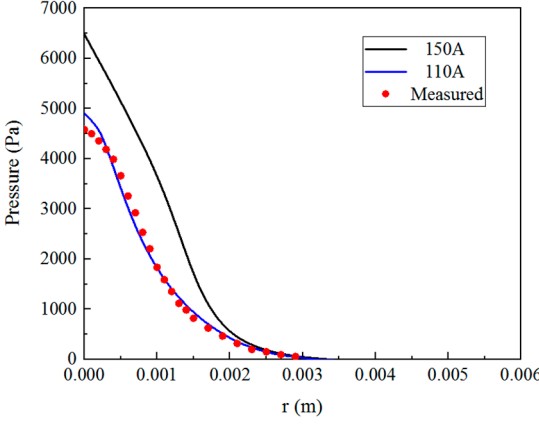

**Figure 13.** Comparison of calculated arc pressure and measured pressure on workpiece surface.

## 5. Conclusions

1.  A convenient unified model was established to display the mobile keyhole-mode arc welding process. It integrates the electro-thermal conversion, electromagnetic effect, heat transfer, solid-liquid phase change, and arc and molten metal flow with a set of multi-physics governing equations. It takes quite less computational time than VOF-based unified models since it simplifies the two-phase flow problem into a single-phase problem.

2.  A high-temperature arc zone appears below the electrode, and it is like an arc column located in the center of the arc. The highest temperature is above 27,000 K adjacent to the electrode tip, while the lowest temperature is 15,000 K at the edge of the arc column. The arc with velocity above 200 m/s almost gathers within a column of diameter of 3 mm, and that below 200 m/s diffuses to 10 mm at the top of workpiece.

3.  The weld pool center and the torch center do not coincide due to the relative movement. A deviation appears, and it may gradually increase to 3.5 mm during the welding process. The weld pool is long at the rear but narrow in the front, and it gradually shrinks from top surface to bottom surface. It shrinks fast at the upper part and smoothly at the lower part, and finally forms a boot-shape weld pool.

4.  The experiment shows the weld pool image and the measured arc pressure agree well with the calculated result. The calculated electric potential drop also coincides with the experiment. The model provides a convenient and accurate method to display the mobile keyhole-mode arc welding process.

**Author Contributions:** Conceptualization, Methodology, Software, Writing—Review and Editing, Funding acquisition, Y.L.; Data curation, Writing—Original draft preparation, C.S.; Software, Formal analysis, Visualization, L.W.; Validation, Investigation, Resources, C.W. All authors have read and agreed to the published version of the manuscript.

**Funding:** This research was funded by National Natural Science Foundation of China (No. 51706246 and No. 52076216).

**Acknowledgments:** The authors thank the China Scholarship Council for providing a scholarship. The authors are grateful to Dr. Zuming Liu for his assistance in arc pressure measurement.

**Conflicts of Interest:** The authors declare no conflict of interest.

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
