# Peer review of "A Convenient Unified Model to Display the Mobile Keyhole-Mode Arc Welding Process"

_applsci, doi:10.3390/app10227955_

Round 1

Reviewer 1 Report

A convenient, unified model has been created to display the arc welding process in the mobile keyhole mode. The new model requires significantly less computational time than the unified VOF-based models because it simplifies the two-phase flow problem to a single-phase one.

The presented approach is of a utilitarian nature. It can be used in industry as it takes much less computational time than previous standardized models. It is very important.

The experiment was properly performed on a stainless steel plate and the image of the weld pool and the measured arc pressure agree with the calculated result. The presented calculations are well described.

I believe that the presented new model provides a convenient and accurate way to display the arc welding process in the mobile buttonhole mode. The article is very interesting, the presented solution can be used in production of welding construction by ARC method. So I think the text will be of interest to readers.

I accept the manuscript after very minor revision :

φw is work function of workpiece and
what is the "φ" without index (table 1) ? Please describe it in text

Reviewer 2 Report

In the subject of the paper is the model of keyhole-mode arc welding process. The keyhole formation during welding is directly connected with formation of high temperature in the center part of weld pool (gas) and the lower temperature around it (only melted metal). So the welding pool consist of two phases – liquid  and gases. Such model of weld pool is presented in fig. 1. Such effect resulted heat and mass transfer in the volume of weld pool, especially in the center zone of molten-gases-state metal.

According to above, in my opinion the whole analysis and results show only the results for melting metal, without it evaporation and gas-phase formation. By removing from the paper the word “keyhole” it is going to be very good paper.

To the methodology of results analysis and to the results I have no additional comments.

Great advantage of paper is the shape of fusion in 304 steel. Considering the process parameters esp. current 110 A, and it good agreement with calculation shows that the welding process could be without keyhole formation. The keyhole formation begins above current of 100 A, but also depends of other process parameters. This test proved that it is to early to considering in model key-hole formation.

Reviewer 3 Report

The article developed a computation efficient unified model to predict the key-hole behaviour od plasma arc welding. The topic is quite relevant to present-day industrial requirements. Few items are to be addressed before publication. The manuscript can be improved by considering the following: 

  • In the introduction, the advantages of PAW are reported. However, authors are requested to mention the disadvantages, issues and challenges as well.
  • In the literature review, authors have reported how other papers modelled the PAW process. However, the outcome and challenges of their methods are to be specified.
  • Which shielding gas was used in this study? Was it Ar? Also, provide the specification of the PAW machine used.
  • Have you done any mesh sensitivity analysis before selecting the mesh size?

Round 2

Reviewer 2 Report

Thank you for your explanation. Of course I accept it.

Summarize, you are considering key-hole mode without key-hole formation by only melting metal. Results as the high pressure in the center of weld pool and the weld pool shape indicate the key-hole formation is possible. Moreover, the experimental data acc. to weld shape and welding parameters (over 100 A) confirm key-hole formation in the welded metal.

All results of modeling analysis indicate only the temperature near to melting point without reaching boiling point. In your’s analysis on the metal side is the temperature only over 1665 K (metal is still solid), where typical melting point for austenitic steel is near to 1800 K and there is not visible are not possible to reached boiling point (look at fig. 5 and 6).